# KD-HGRL: Knowledge Distillation for Multi-Task Heterogeneous Graph Representation Learning

## Abstract

Heterogeneous graphs, characterized by diverse node and edge types, are central to many real-world applications, including social networks, biological systems, and recommendation engines. While Graph Neural Networks (GNNs) are effective for graph representation learning, their reliance on extensive labeled data, high computational cost, and long inference times limit scalability, especially for heterogeneous graphs. To address these challenges, we propose KD-HGRL, which leverages **K**nowledge **D**istillation for multi-task **H**eterogenous **G**raph **R**epresentation **L**earning. KD-HGRL uses self-supervised contrastive learning across semantic and topological views to generate robust, label-free node embeddings in the teacher phase. These embeddings are distilled into a lightweight student model, enabling efficient task-specific outputs such as node classification and link prediction with significantly reduced inference time. Experiments on benchmark datasets demonstrate KD-HGRL's superior performance and efficiency compared to state-of-the-art methods. The framework captures both local and global graph structures, eliminates the need for labeled data, and scales effectively to large graphs. Key novelties, such as a multi-view teacher model, contrastive alignment, and a lightweight student model, make KD-HGRL a versatile and efficient solution for heterogeneous graph representation learning.

## 1 Introduction

Graphs are essential for modelling complex relationships in various domains, including social networks Wasserman & Faust (1994), biological systems Pavlopoulos et al. (2011), and recommendation engines Pavlopoulos et al. (2011). Some real-world networked systems feature diverse node and edge types, such as bibliographic networks with authors, papers, and venues, referred to as heterogeneous graphs Wang et al. (2022b). Graph representation learning is crucial for encoding graph data into vectors that preserve key properties of the graph, such as node relationships, network topology, and feature information Hamilton (2020). Graph Neural Networks (GNNs) have proven effective in learning graph representations, transforming nodes, edges, or entire graphs into low-dimensional vectors while preserving their structural relationships Zhang et al. (2019a). However, GNNs face critical challenges, especially in supervised learning, where their performance is highly dependent on having access to large amounts of labeled data. Labeled data is often limited or costly in real-world applications, making it difficult for GNNs to effectively learn the complex relationships between nodes, edges, and their features. This reliance on extensive labeled data limits their scalability and effectiveness in scenarios where only small or incomplete labeled data is available, leading to the need for alternative methods like semi-supervised Wan et al. (2021), meta-learning Ding et al. (2022), and transfer learning Zhu et al. (2024) approaches to mitigate these challenges Khemani et al. (2024). However, they still require a substantial amount of labeled nodes in each class to achieve satisfactory results.

**Motivation**: One effective approach to mitigate these challenges is knowledge distillation (KD), where a pre-trained teacher model transfers knowledge to a student model, allowing the student to perform well even with limited labeled data Tian et al. (2023). Existing studies on knowledge distillation, such as Shen et al. (2025), focus on homogeneous graphs, overlooking the complexities

arising from the diverse node types and edges in heterogeneous graphs. Some works, such as Wang et al. (2022a) and Fu et al. (2024), employ collaborative knowledge transfer to extract node embeddings from heterogeneous graphs. However, these methods are limited to node classification tasks. Their approach used a knowledge transfer strategy across various meta-paths to enhance the quality of embeddings instead of transferring knowledge between two models. Some approaches, such as those proposed in Liu et al. (2022a), Zhang et al. (2022), and Feng et al. (2024), use soft labels to transfer knowledge from teacher models to student models. In these methods, the teacher model is trained in a supervised setting for a specific task, making the approach task-dependent and unsuitable for transferring knowledge to various downstream tasks such as node classification, link prediction, and influence maximization, which may have conflicting objectives or require different representations. Furthermore, using soft labels fails to capture the complex high-order structures of the graph learned by the teacher model. Moreover, most GNN-based approaches for heterogeneous graph embedding, such as Ma et al. (2023), require substantial memory and computational resources during inference, particularly when processing large-scale graphs. This is primarily due to the reliance on message passing, where each node aggregates information from its neighbours across multiple layers. As the size of the graph grows, the computational cost increases exponentially with the depth of the network, making it particularly challenging to scale these methods to large, real-world graphs. For instance, if the average number of neighbors is $R$, a GNN with $L$ layers requires approximately $O(R^L)$ computations to extract the embeddings of a target node. To improve inference time, some approaches, such as Zhang et al. (2022) and Zhang et al. (2022), transfer knowledge from a teacher model to an MLP as the student model. While MLPs offer fast inference times, they cannot capture the high-order and complete structural information learned by the teacher model. Moreover, these models rely on soft labels to transfer knowledge, which are task-dependent and not well-suited for various downstream tasks. Thus, two key questions arise, *Q1: How can we develop an effective teacher model tailored for heterogeneous graphs that leverages self-supervised learning and is applicable to multiple downstream tasks?* and *Q2: How can we design a lightweight student model that ensures fast inference time while capturing the high-order relationships of the graph learned by the teacher?*

**Present work**: To mitigate these challenges, we propose KD-HGRL (Knowledge Distillation for Multi-Task Heterogeneous Graph Representation Learning). This novel framework leverages knowledge distillation (KD) to address limited labelled data and inference time issues. In the teacher phase, we use self-supervised contrastive learning with two distinct views of the heterogeneous graph: semantic and topological. The semantic view captures node embeddings via a Graph Convolutional Network (GCN) to learn topological relationships. In contrast, the topological view uses node features to generate global representations, enriching the embeddings with topological information. A contrastive loss function aligns the embeddings from both views, generating robust node representations without requiring label data. This phase is highly efficient as it does not rely on labelled nodes or links, making it scalable and adaptable for large-scale heterogeneous graphs. In the student phase, knowledge distillation transfers the learned knowledge from the teacher model to a lightweight student model. The student model is based on a lightweight GCN combined with a MLP, designed to predict task-specific outputs such as node classification and link prediction. The student model learns from the teacher's representations of the nodes and their neighbors, reducing the inference time significantly compared to the teacher model, which processes the entire graph with deeper layers and higher compexity. The student model uses only the subgraph of each node and its first-hop neighbours, making it computationally more efficient while still maintaining performance. This method improves the efficiency of inference and allows for multi-task learning to handle tasks like node classification and link prediction simultaneously. Experiments on real-world benchmark datasets demonstrate that KD-HGRL outperforms state-of-the-art approaches, balancing high performance and reduced computational cost.

**Novelties**: The proposed framework, KD-HGRL, introduces several key novelties that address critical challenges in heterogeneous graph representation learning and knowledge distillation. These are **(1)** Unlike existing knowledge distillation, approaches that rely on supervised learning and task-specific labels Liu et al. (2022a); Zhang et al. (2022); Shen et al. (2025), our framework employs self-supervised contrastive learning to train the teacher model. This eliminates the dependency on labeled data, making it applicable to large-scale heterogeneous graphs where labels are scarce or unavailable. **(2)** Unlike methods such as Wang et al. (2022a); Fu et al. (2024); Wang et al. (2021c), which use only meta-paths to extract representations of heterogeneous graphs, our method leverages two distinct views (e.g. semantic and topological) the teacher model captures both the structural

and semantic information of the graph, aligning them using a contrastive loss to generate robust node embeddings. **(3)** Previous methods, such as Liu et al. (2022a) and Feng et al. (2024), transfer knowledge using soft labels, which fail to preserve the structural properties of the graph. In contrast, KD-HGRL distills rich multi-view representations from the teacher to a lightweight student model, effectively integrating local and global graph information. **(4)** Compared to methods such as Feng et al. (2024), which use only an MLP as the student model, KD-HGRL introduces a computationally efficient student model called LightGCN. This model uses just one layer, combined with an MLP, to capture the high-order relationships of the graph learned by the teacher model. This mechanism balances performance and computational efficiency effectively, making it well-suited for real-world applications with limited computational resources.

## 2 PRELIMINARY

**Heterogeneous graph (HG)** Zhang et al. (2019b): A HG is a graph where nodes and/or edges belong to multiple types, making it more expressive for representing complex systems. Formally, an HG can be defined as $G = (V, E, \phi, \psi)$, where $V$ is the set of nodes, $E \subseteq V \times V$ is the set of edges, and $\phi : V \to A$ is the node type mapping function. $A$ is the set of node types and $\psi : E \to R$ is the edge type mapping function, where $R$ is the set of edge types. In this formulation, each node $v \in V$ is associated with a type $\phi(v) \in A$, and each edge $e \in E$ is associated with a type $\psi(e) \in R$. A meta-path in a HG is defined as a sequence of relations between different node types, denoted as $\Gamma_1 \xrightarrow{r_1} \Gamma_2 \xrightarrow{r_2} \dots \xrightarrow{r_l} \Gamma_{l+1}$, where each relation $r_i \in R$ represents a specific edge type. Meta-paths describe composite relations between two node types, capturing the structural and semantic relationships within the graph.

**Knowledge Distillation** Phuong & Lampert (2019): Given a teacher model $f_T$ and a student model $f_S$, KD aims to transfer the knowledge from the teacher to the student by aligning the student model's predictions with the teacher's output. This process is quantified through mutual information maximization. Formally, it can be expressed as minimizing the Kullback-Leibler (KL) divergence between the teacher's prediction $p_T(y|x)$ and the student's prediction $p_S(y|x)$, represented as:

$$\min KL(p_T(y|x) \parallel p_S(y|x)) = \min \sum_x \sum_y p_T(y|x) \log \frac{p_T(y|x)}{p_S(y|x)} \tag{1}$$

This ensures that the student model closely approximates the behavior of the teacher model.

**Graph Convolutional Network**: GCN Jin et al. (2021) is an effective model for learning node embeddings by incorporating both the graph structure $G$ and node feature matrix $X$. In this paper, GCN is utilized as the encoder to compute node embeddings $h_i \in \mathbb{R}^d$ (where $d$ is the embedding dimension) for each node $v_i$. In GCN the update rule for propagating the representations at different layers is defined as:

$$H^{(l+1)} = \sigma(\tilde{A} H^{(l)} W^{(l)}) \tag{2}$$

Here, $H^{(l+1)}$ is the embeddings of the node at the $(l+1)$-th layer, while $H^{(0)} = X$ represents the initial node features. The matrix $\tilde{A}$ is the normalized adjacency matrix with self-loops, calculated as $\tilde{A} = \hat{D}^{-\frac{1}{2}} \hat{A} \hat{D}^{-\frac{1}{2}}$, where $\hat{A} = A + I_N$ (with $A$ and $I_N$ denote the adjacency matrix and the identity matrix, respectively) and $\hat{D}$ denotes the diagonal degree matrix of $\hat{A}$. The matrix $W^{(l)}$ represents the weight matrix at the $l$-th layer, and $\sigma$ denotes the activation function. For simplicity, we denote the GCN model as $z = \text{GCN}(X, A)$, where $z$ represents the resulting node embeddings.

## 3 PROPOSED METHOD

This paper introduces a novel **K**nowledge **D**istillation framework for multi-task **H**eterogenous **G**raph **R**epresentation **L**earning called KD-HGRL. The proposed framework is based on the teacher-student KD paradigm, where the teacher model aims to generate rich and comprehensive node embeddings by leveraging multi-view learning from the heterogeneous graph. The student model, in turn, is designed to learn from these embeddings while being more lightweight and computationally efficient. The goal is to transfer the knowledge captured by the teacher to the student, ensuring that the student can achieve high performance on multiple tasks, such as node classification and link prediction, with lower computational costs. The overall structure of the proposed method is presented in 1

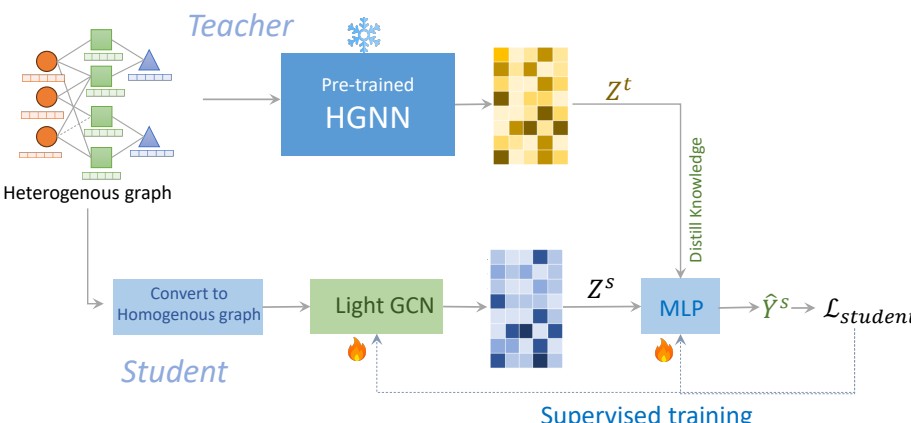

Figure 1: The proposed KD-HGRL framework consists of two main components: (a) A teacher model that utilizes a pre-trained, multi-view, contrastive learning-based graph neural network. The teacher computes rich node embeddings from both semantic and topological views, as detailed in Figure 2 (b) A student model that utilizes the embeddings transferred from the teacher into a more compact and efficient model, optimized for downstream tasks such as node classification and link prediction

### 3.1 TEACHER PHASE: SELF-SUPERVISED CONTRASTIVE LEARNING ON HETEROGENEOUS GRAPHS

The teacher phase employs two complementary views: the semantic and topological views. In the semantic view, meta-paths capture complex semantic relationships between nodes, generating homogeneous graphs for each path. A GCN is applied to these graphs to learn node embeddings, and since each node type can have multiple meta-paths, an attention mechanism integrates the embeddings for each node type across these different paths. The topological view operates on the heterogeneous graph, using a message-passing GCN to allow nodes to exchange information with neighbors of different types. This approach aggregates features from the local neighborhood, generating initial node embeddings. Afterward, we build a global graph where nodes are linked based on their immediate neighbors and similarities, even if they are not directly connected in the original graph. This global graph captures higher-order structural information. A contrastive loss is used to align the node representations learned from both views, ensuring consistency between the embeddings from the semantic and topological perspectives. By maximizing the similarity between positive pairs (nodes with shared features or class) and minimizing the similarity for negative pairs (unrelated or unconnected nodes), the model learns robust and discriminative representations that combine both semantic and structural knowledge. The overall structure of the teacher phase, including the semantic and topological views, is illustrated in Figure 2. **Semantic view:** Let $C$ represent the set of node types. For each node type $c \in C$, we define a set of meta-paths $M^c$. For each meta-path $m_i^c \in M^c$ (i.e., $M^c = \{m_1^c, m_2^c, \ldots, m_k^c\}$), a corresponding homogeneous graph $g_i^c$ is generated, represented by an adjacency matrix $A_i^c$. To account for self-loops, we define the adjusted adjacency matrix $\tilde{A}_i^c = A_i^c + I^c$, where $I^c$ is the identity matrix for node type $c$. Each graph is also associated with a degree matrix $\tilde{D}_i^c$, where the diagonal entries represent node degrees. We then apply a GCN to each meta-path-induced homogeneous graph. The node representations at the $(l+1)$-th layer for graph $g_i^c$, denoted by $H_i^{(c,l+1)}$, are computed using the GCN update rule:

$$H_i^{(c,l+1)} = \delta \left( (\tilde{D}_i^c)^{-1/2} \tilde{A}_i^c (\tilde{D}_i^c)^{-1/2} H_i^{(c,l)} W^{(l)} \right) \tag{3}$$

where $\delta(\cdot)$ represents a non-linear activation function (e.g., ReLU), $H_i^{(c,l)}$ is the node embedding matrix at layer $l$, and $W^{(l)}$ is the weight matrix for layer $l$. $W^{(l)}$ is the trainable weight matrix at layer $l$, and $\delta(\cdot)$ is the ReLU activation function defined as $\delta(x) = \max(0, x)$. If the graph nodes have predefined features (such as attributes or descriptors), these can be directly used as the initial

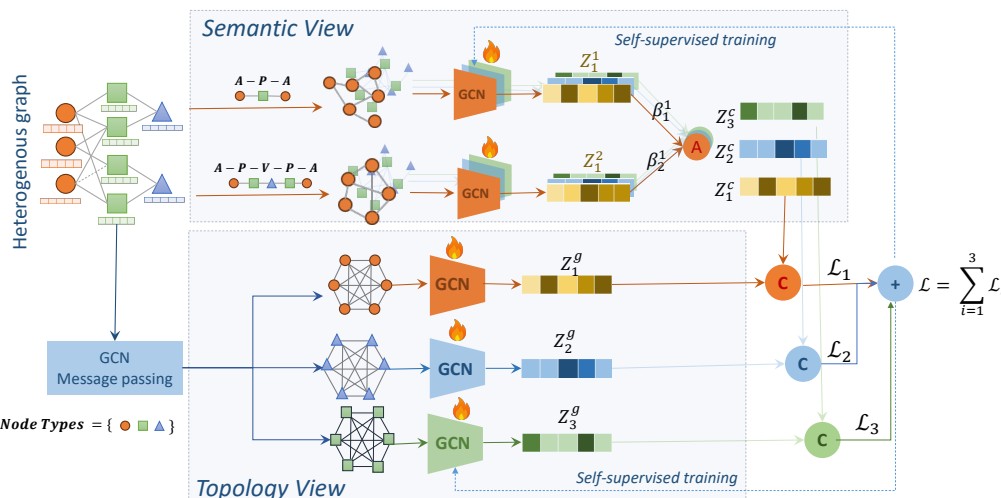

Figure 2: The overall architecture of the teacher model.

node representations $Z_i^{(c,0)} = X^c$, where $X^c$ corresponds to the features of node type $c$. Without node features, one-hot encoding can be used as $Z_i^{(c,0)} = I^c$. After obtaining node embeddings from different meta-paths, we use an attention mechanism to aggregate these embeddings. The attention weight $\beta_i^c$ for each meta-path is computed based on the importance of its corresponding node representation. One common approach is to learn $\beta_i^c$ through softmax normalization as:

$$\beta_i^c = \frac{\exp(\text{score}(Z_i^c))}{\sum_{j \in C} \exp(\text{score}(Z_j^c))} \tag{4}$$

where $Z_i^c = H_i^{(c,L)}$ is the final node representation from the GCN after $L$ layers, and $\text{score}(Z_i^c)$ is a learned scalar score that measures the importance of meta-path $m_i^c$ for node type $c$. The final node embedding for node type $c$ is obtained by aggregating the embeddings from different meta-paths, weighted by their respective attention scores $\beta_i^c$ for each meta-path $m_i^c$, learned during training. The aggregated representation for each node type $c$ is computed as:

$$Z^c = \sigma \left( \sum_{i=1}^k \beta_i^c Z_i^c \right) \tag{5}$$

where $Z_i^c = H_i^{(c,L)}$ is the final node representation from the GCN after $L$ layers, and $\sigma(\cdot)$ is a non-linear function like ReLU or softmax.

**Topological view:** This view utilizes a novel strategy to derive a global representation of the graph from the node features. Let $X$ denote the node features, where $X_i$ corresponds to the feature vector of node $n_i$. We first obtain a representation for each node $n_i$, denoted as $Z_i^f$:

$$Z_i^f = \frac{1}{|R_i| + 1} \sum_{r \in R_i} \left( \sum_{j \in N_i(r)} \frac{1}{|N_i(r)|} W_r X_j + W_0 X_i \right) \tag{6}$$

where $R_i$ denotes the set of node types that node $n_i$ is connected to, $N_i(r)$ is the set of nodes of type $r$ connected to $n_i$, $W_r$ is the learnable weight matrix for node type $r$, $W_0$ is the weight matrix for node $n_i$, and $X_i$ and $X_j$ are the features of nodes $n_i$ and $n_j$, respectively. The final embedding for each node is computed by aggregating the information from its neighbors as:

$$Z_i^g = \sigma \left( \sum_{\forall j} f(Z_i^f, Z_j^f) Z_j^f \right) \tag{7}$$

where $f(Z_i^f, Z_j^f)$ represents an attention score between $n_i$ and $n_j$, which is defined as:

$$f(Z_i^f, Z_j^f) = \frac{\exp(\text{LeakyReLU}(Z_i^f \parallel Z_j^f))}{\sum_{\forall k} \text{LeakyReLU}(Z_i^f \parallel Z_k^f)} \tag{8}$$

Here, $\parallel$ denotes the concatenation of the representations, and $\sigma(\cdot)$ is a non-linear activation function like ReLU.

**Self-supervised learning:** We use a contrastive learning approach to combine the meta-path-based embedding $Z^c$ and the global semantic embedding $Z^g$. The contrastive learning framework aims to align these two representations for the same node while distinguishing them from the embeddings of other nodes. To achieve this, we define a contrastive loss function based on the normalized temperature-scaled cross-entropy loss. The loss is designed to maximize the cosine similarity between the embeddings $Z_i^c$ and $Z_i^g$ for the same node $n_i$ while minimizing the similarity between the embeddings of different nodes. The contrastive loss for a positive pair $Z_i^c$ and $Z_i^g$ is defined as:

$$L_i = -\log\left(\frac{\exp(\text{sim}(Z_i^c, Z_i^g))}{\sum_{j=1, j \neq i}^{N} \exp(\text{sim}(Z_i^c, Z_j^g))}\right) \tag{9}$$

where $\tau$ is a temperature scaling parameter, $N$ is the total number of nodes, and $\text{sim}(\cdot, \cdot)$ calculates the similarity between two embeddings. The final loss function is computed as the summation of the loss function for all nodes and is defined as:

$$L = \frac{1}{N} \sum_{i=1}^{N} L_i \tag{10}$$

By minimizing this contrastive loss $L$, this method effectively combines the structural information from the meta-path view with the semantic information from the global view, leading to richer node embeddings in heterogeneous graphs.

## 3.2 KNOWLEDGE DISTILLATION FOR NODE CLASSIFICATION

This section explains how the proposed framework is applied to the node classification task as shown in Figure 3. The student model for this task is designed to predict the class of nodes using a combination of a lightweight GCN and an MLP. To this end, a homogeneous graph is first generated for the target node type and then fed into the GCN to generate the node embeddings. For each node, the teacher does not simply pass the embedding of the target node alone to transfer the knowledge. Instead, it calculates the average of the embeddings from both the target node and its class-specific neighbors. The embedding transferred from the teacher is later combined with the

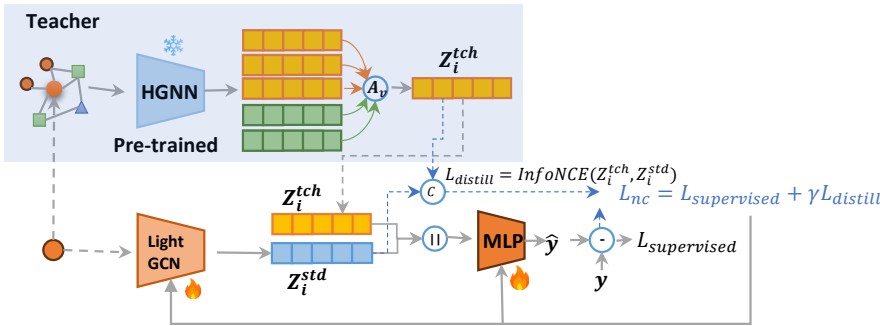

Figure 3: Overview of the node classification framework using KD-HGRL, where the student model combines embeddings from a lightweight GCN and the teacher model. The teacher integrates local structure and global class centroids, facilitating knowledge transfer through supervised and distillation losses for efficient classification.

task-specific embedding generated by the lightweight GCN. The combined embedding is then fed

into the MLP to generate the task-specific output. The student model is trained using labeled data with a supervised loss function. Let $Z_i^{\text{std}}$ denote the embedding generated for a target node $n_i$ by aggregating information from its first-hop neighbours. This is achieved through the standard GCN update rule, limited to a single meta-path $m_1$:

$$Z_i^{\text{std}} = \delta \left( \sum_{j \in N_i} \frac{1}{\sqrt{d_i d_j}} A_{ij} Z_j W^{\text{std}} \right) \tag{11}$$

where $N_i$ represents the neighbors of node $n_i$, $A_{ij}$ is the adjacency matrix, $d_i$ is the degree of node $n_i$, $W^{\text{std}}$ is a trainable weight matrix, and $\delta(\cdot)$ is a non-linear activation function such as ReLU. The teacher model transfers knowledge to the student model by integrating local structural information from a node's subgraph with global semantic information from class centroids. These centroids represent the average embeddings of nodes within a specific class, enriching the target node's representation during the knowledge transfer process. To compute the class centroids $P_{c_i}$ for each class $c_i$, the teacher model averages the embeddings $Z_j^{\text{tch}}$ for nodes within subgraphs related to that class:

$$P_{c_i} = \frac{1}{|V_{c_i}|} \sum_{n_j \in V_{c_i}} Z_j^{\text{tch}} \tag{12}$$

where $V_{c_i}$ is the set of nodes classified as belonging to class $c_i \in C$, and $Z_j^{\text{tch}}$ is the teacher embedding for node $n_j$. To derive the final teacher embedding for a target node $n_i$, we combine the local node embeddings from the subgraph $S_i$ and the global class-level centroids $P_{c_i}$. This fusion incorporates both neighborhood information and class knowledge. The final teacher embedding $Z_i^{\text{tch}}$ is given by:

$$Z_i^{\text{tch}} = \alpha \sum_{n_j \in S_i} w_{ij} Z_j^{\text{tch}} + (1 - \alpha) \sum_{c_k \in C} \text{softmax}(c_k)(\text{sim}(Z_i^{\text{tch}}, P_{c_k})) P_{c_k} \tag{13}$$

where $w_{ij}$ is the weight representing the importance of node $n_j$ in subgraph $S_i$, calculated using a similarity function (e.g., cosine similarity). $\alpha \in [0, 1]$ is a parameter balancing the influence of the subgraph embeddings and class centroids. $\text{sim}(Z_i^{\text{tch}}, P_c)$ denotes the similarity between the node's teacher embedding and the class centroid $P_c$. The softmax function is used to compute the likelihood of the node $n_i$ belonging to each class $c$, based on the similarity of its embedding to each class centroid. The prediction for node $n_i$, denoted as $\hat{y}_i$, is obtained by combining the student's embedding $Z_i^{\text{std}}$ and the teacher's embedding $Z_i^{\text{tch}}$ using a Multi-Layer Perceptron (MLP):

$$\hat{y}_i = \text{MLP}(Z_i^{\text{std}} \parallel Z_i^{\text{tch}}) \tag{14}$$

Here, $\hat{y}_i$ is a vector that includes the probability of the node belonging to each class, and $\parallel$ denotes the concatenation operation. This approach ensures that both the student and teacher models contribute to the final node classification. The student model is trained using two losses. First, a supervised loss ensures that the model's predictions align with the true class labels. The supervised loss, based on cross-entropy, is calculated as:

$$L_{\text{bpr}} = - \sum_{i=1}^{N} \sum_{c=1}^{C} y_{i,c} \log(\hat{y}_{i,c}) \tag{15}$$

where $y_{i,c}$ is the true label for node $i$ and $\hat{y}_{i,c}$ is the predicted probability for class $c$. This ensures the model focuses on correctly classifying nodes. Additionally, a distillation loss aligns the student representation with the teacher. To ensure that the light GCN in the student model captures similar high-order relationships as the HGNN, a contrastive learning approach is applied using the InfoNCE loss:

$$L_{\text{distill}} = - \frac{1}{|V|} \sum_{v \in V} \log \left( \frac{\exp \left( Z_v^{\text{std}} \cdot Z_v^{\text{tch}} / \tau \right)}{\sum_{v' \in V} \exp \left( Z_{v'}^{\text{std}} \cdot Z_{v'}^{\text{tch}} / \tau \right)} \right) \tag{16}$$

where $\tau$ is the temperature parameter that scales the similarity scores between embeddings. The total loss for the teacher model combines the supervised and contrastive losses as:

$$L_{\text{nc}} = L_{\text{bpr}} + \gamma L_{\text{distill}} \tag{17}$$

with $\gamma$ as a hyperparameter balancing the contributions of both losses. This ensures a seamless transfer of the topological structure and high-order information from the HGNN to the light GCN.

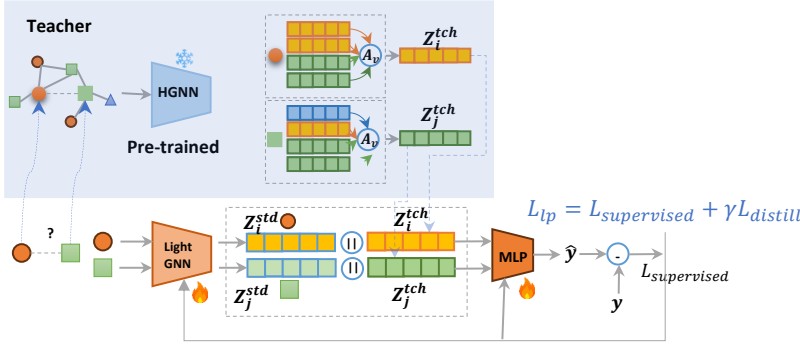

Figure 4: Overview of the KD-HGRL link prediction framework. The model combines teacher and student embeddings through an attention mechanism, predicting links with a multi-layer perceptron (MLP) while using supervised and distillation losses for accuracy and knowledge transfer.

### 3.3 KNOWLEDGE DISTILLATION FOR LINK PREDICTION

In our approach to link prediction, we extend the KD framework to predict the likelihood of a link between two nodes $i$ and $j$. This involves deriving embeddings from both teacher and student models based on their respective learned representations. For each node $i$ and $j$, we first extract subgraphs centered around these nodes to compute their embeddings. The embeddings from the teacher model are represented as $Z_i^{\text{tch}}$ and $Z_j^{\text{tch}}$, while the embeddings from the student model are denoted as $Z_i^{\text{std}}$ and $Z_j^{\text{std}}$. To combine the embeddings from both phases, we utilize an attention mechanism. The final embedding for node $i$ and $j$ is computed as:

$$Z_i = \text{MLP}\left(\text{softmax}(W \cdot [Z_i^{\text{tch}}, Z_i^{\text{std}}])\right) \tag{18}$$

$$Z_j = \text{MLP}\left(\text{softmax}(W \cdot [Z_j^{\text{tch}}, Z_j^{\text{std}}])\right) \tag{19}$$

Here, $W$ is a weight matrix, and the softmax function produces attention weights for the respective embeddings, allowing the model to learn the importance of each representation. Once we have the combined embeddings for nodes $i$ and $j$, we predict the probability of a link between them using an MLP as

$$\hat{y}_{ij} = \text{MLP}(Z_i, Z_j) \tag{20}$$

We define two loss functions in our framework: the supervised loss and the distillation loss. The supervised loss, $L_{\text{supervised}}$, ensures that the predicted probability $\hat{y}_{ij}$ matches the true label $y_{ij}$, which indicates whether a link exists between nodes $i$ and $j$. It is defined as:

$$L_{\text{supervised}} = \sum_{(i,j) \in D} \text{BCE}(\hat{y}_{ij}, y_{ij}) \tag{21}$$

where $\text{BCE}(a, b)$ denotes the binary cross-entropy between the predicted link probability $\hat{y}_{ij}$ and the true label $y_{ij}$. Also, To transfer knowledge from the teacher model to the student model, we define a distillation loss, $L_{\text{distill}}$, which encourages the student model to approximate the teacher's predictions. The distillation loss is formulated as:

$$L_{\text{distill}} = \sum_{(i,j) \in D} \left(\lambda \cdot \text{BCE}(\sigma(Z_i^{\text{tch}} \cdot Z_j^{\text{tch}}), y_{ij}) + (1 - \lambda) \cdot \text{BCE}(\sigma(Z_i^{\text{std}} \cdot Z_j^{\text{std}}), y_{ij})\right) \tag{22}$$

In this equation, $\sigma(\cdot)$ denotes the sigmoid function, and the dot product between the embeddings $Z_i^{\text{tch}}$ and $Z_j^{\text{tch}}$ (and similarly for the student embeddings) reflects the predicted link probability from the respective model. The weights $\lambda_1$ and $\lambda_2$ balance the contributions of the teacher and student models. The total loss for training the student model is similar to the node classification part, as shown in equation (17). This formulation encourages the student model to learn effective link prediction capabilities by distilling knowledge from the teacher model while ensuring accurate predictions based on the final combined embeddings. Note that the loss function varies for each task. Specifically, we use different loss functions for node classification and link prediction.

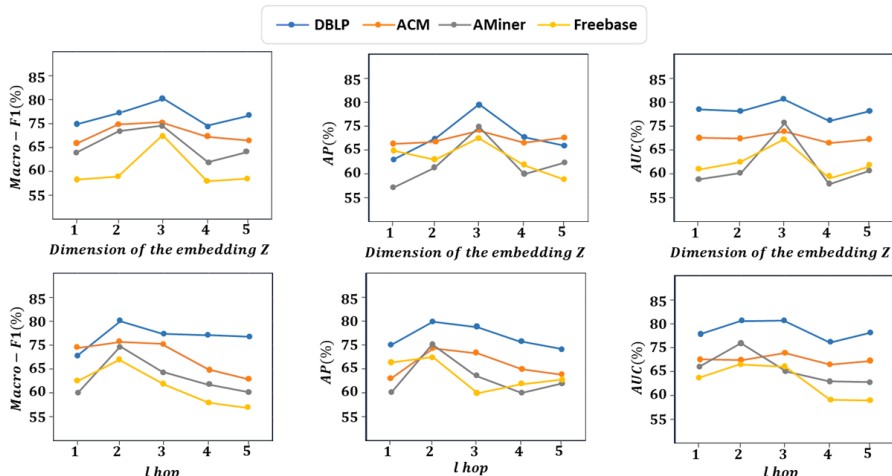

Figure 5: Performance of KD-HGRL with varying embedding dimensions and hop counts.

## 4 EXPERIMENTS

**Datasets:** The datasets used in this study encompass a diverse range of domains and relationships. Freebase Li et al. (2021), ACM Zhang et al. (2019a), DBLP and Freebase Fu et al. (2020). These datasets are commonly used for Benchmarking tasks in heterogeneous graph learning studies, such as node classification and link prediction.

**Baseline Methods:** We compare KD-HGRL against five notable network representation learning methods, including two supervised approaches: HAN Wang et al. (2021a) and MAGNN Fu et al. (2020); two unsupervised (self-supervised) methods: HeCo Wang et al. (2021c) and HeMue Zhang et al. (2023); and one model utilizing fine-tuning based on meta and prompt learning for heterogeneous graphs, HetGPT.

**Implementation and Parameter Settings** Experiments were conducted 10 times, averaging results across datasets for fairness. The embedding dimension was fixed at 64, using original attributes for target nodes and one-hot encoding for others as needed. KD-HGRL employed Glorot initialization Glorot & Bengio (2010), the Adam optimizer Kingma & Ba (2014) (learning rates: 1e-4 to 5e-3), early stopping (patience: 5-50), dropout (0.1-0.5), and $\tau$ (0.5-0.9). For knowledge distillation, pre-trained teacher model parameters were frozen to retain embeddings and structural insights, while a learnable feature vector enhanced representation capabilities. The student model was fine-tuned during downstream training to optimize knowledge transfer and task performance. Performance analysis (Fig. 3) shows the model peaked at 64-dimensional embeddings, with redundancy lowering effectiveness at higher dimensions. Additionally, 2-hop neighborhoods yielded the best results, with performance declining as additional hops introduced redundancy and complexity, reducing effectiveness in heterogeneous data management.

**Evaluating the Node Classification Task:** We assess the node classification performance using two data splits, (80%, 10%, 10%) and (60%, 20%, 20%). As displayed in Table 2, KD-HGRL and HetGPT outperform other models across all datasets. KD-HGRL stands out with a Micro-F1 of 87.34 on DBLP, significantly surpassing the other methods, while HetGPT also delivers strong results, particularly on Freebase and ACM. Both models exhibit superior predictive power, with KD-HGRL demonstrating effective generalization due to its contrastive learning approach, which captures both structural and semantic features. Results for the 6/2/2 split and additional label ratio evaluations are detailed in Appendix A.3, further validating the robustness of KD-HGRL.

**Link Prediction Evaluation:** We evaluated KD-HGRL using two edge splits: 80%/10%/10% for training, validation, and testing. Table 2 presents the results for the 80%/10%/10% split, where KD-HGRL and HetGPT outperform other models (MAGNN, HAN, HeCO, HeMuc) across all datasets.

Table 1: Node Classification

| Models | Datasets | Metrics | MAGNN | HAN | HeCO | HeMuc | HetGPT | KD-HGRL |
|---|---|---|---|---|---|---|---|---|
| Split (80%, 10%, 10%) | Freebase | Micro_F1 | 64.13 | 64.27 | 68.42 | 69.35 | 75.89 | **78.34** |
| | | AP | 63.45 | 63.89 | 67.15 | 68.01 | 74.34 | **76.12** |
| | | AUC | 62.34 | 62.77 | 66.08 | 66.94 | 73.12 | **75.87** |
| | ACM | Micro_F1 | 65.84 | 66.09 | 70.29 | 71.55 | 80.13 | **82.76** |
| | | AP | 64.67 | 64.92 | 69.56 | 70.70 | 78.78 | **80.34** |
| | | AUC | 63.89 | 64.11 | 68.66 | 69.45 | 77.23 | **79.67** |
| | AMiner | Micro_F1 | 64.58 | 64.71 | 68.65 | 70.02 | 76.13 | **78.50** |
| | | AP | 63.21 | 63.50 | 67.31 | 68.14 | 75.19 | **77.88** |
| | | AUC | 61.90 | 62.12 | 66.11 | 66.98 | 73.55 | **75.67** |
| | DBLP | Micro_F1 | 66.12 | 66.35 | 70.95 | 72.89 | 85.45 | **92.83** |
| | | AP | 65.23 | 65.50 | 69.74 | 70.59 | 83.10 | **89.24** |
| | | AUC | 63.79 | 64.10 | 67.82 | 68.67 | 80.55 | **88.34** |

Notably, KD-HGRL achieves a Micro-F1 score of 87.34 on DBLP, while HetGPT shows strong performance, particularly on Freebase and ACM. Detailed results for the second split is available in Appendix A.3, demonstrating KD-HGRL's robustness across different configurations.

Table 2: Link prediction split (80%/10%/10%)

| Models | Datasets | Metrics | MAGNN | HAN | HeCO | HeMuc | HetGPT | KD-HGRL |
|---|---|---|---|---|---|---|---|---|
| Split (80%, 10%, 10%) | Freebase | Micro_F1 | 59.25 | 60.19 | 62.89 | 64.24 | 70.42 | **72.35** |
| | | AP | 58.34 | 58.76 | 61.32 | 62.85 | 68.77 | **71.09** |
| | | AUC | 57.23 | 57.95 | 60.19 | 61.44 | 67.89 | **70.12** |
| | ACM | Micro_F1 | 60.12 | 61.45 | 65.39 | 66.88 | 74.29 | **77.03** |
| | | AP | 59.24 | 60.34 | 63.88 | 65.02 | 72.56 | **74.89** |
| | | AUC | 58.55 | 59.12 | 62.45 | 63.68 | 70.73 | **73.87** |
| | AMiner | Micro_F1 | 58.67 | 59.12 | 63.41 | 64.15 | 70.91 | **73.02** |
| | | AP | 57.44 | 58.34 | 61.89 | 62.78 | 68.95 | **71.45** |
| | | AUC | 56.89 | 57.23 | 60.67 | 61.74 | 67.14 | **69.98** |
| | DBLP | Micro_F1 | 61.23 | 61.88 | 67.39 | 69.14 | 80.57 | **87.34** |
| | | AP | 60.35 | 60.78 | 65.84 | 66.55 | 78.21 | **84.23** |
| | | AUC | 58.87 | 59.11 | 64.23 | 64.98 | 76.34 | **83.15** |

## 5 CONCLUSION

In this work, we introduce KD-HGRL, a pioneering framework that leverages knowledge distillation for multi-task learning in heterogeneous graph representation. Our approach addresses critical challenges in the field, such as limited labeled data and high inference costs, by utilizing a self-supervised teacher model that learns from both semantic and topological perspectives of the graph. The teacher's rich embeddings are distilled into a computationally efficient student model, significantly reducing inference time while maintaining high performance on tasks like node classification and link prediction. Experimental results show that KD-HGRL outperforms leading methods, including HAN, MAGNN, and HetGPT, on benchmark datasets such as Freebase, ACM, and DBLP, demonstrating its ability to produce robust embeddings without requiring labeled data. By combining contrastive learning with multi-view graph representations, KD-HGRL offers a scalable and efficient solution for real-world applications. Future work will focus on extending the framework to dynamic graphs, exploring hypergraphs for higher-order relationships, and applying the model to diverse domains such as recommendation systems and fraud detection, further establishing its versatility and effectiveness.

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

# A   APPENDIX

## A.1   RELATED WORKS

**Pre-training Techniques in Graph Learning:** In graph learning, the concept of pre-training is inspired by the remarkable success of pre-trained models in computer vision and natural language processing domains Hu et al. (2019). It involves self-supervised learning on unlabeled graphs to capture their inherent properties. The representations learned during pre-training are not specific to any particular tasks; they are later fine-tuned for specific downstream tasks Lu et al. (2021); Qiu et al. (2020); Hu et al. (2020). Heterogeneous graph pre-training has garnered significant attention, where labeled nodes are often scarce Liu et al. (2022b); Jiang et al. (2021a). The pre-training techniques for learning graphs are classified into generative and contrastive methods. Generative methods aims to reconstruct the graph segments to capture underlying structures or specific node attributes Hu et al. (2020); Jiang et al. (2021b); Yuan et al. (2024); Ren et al. (2023). In contrast, contrastive methods focus on learning representations by maximizing the similarity between positive pairs (nodes expected to be similar or connected, such as those from the same class ) and minimizing it for negative pairs (those expected to be dissimilar or unconnected) Jiang et al. (2021b).Some methods focus on contrasting node-level representations Yang et al. (2022); Wu et al. (2021); Wang et al. (2021b), while others contrast node-level and graph-level representations simultaneously Jing et al. (2021); Park et al. (2020); Ren et al. (2019); Zhou et al. (2022). This approach often proves more effective than generative methods Wang et al. (2023), making it a favored pre-training strategy in heterogeneous graph learning.

**Knowledge Distillation Methods:** KD involves transferring knowledge from a large, complex model (the teacher) to a smaller, simpler model (the student) to reduce computational costs while maintaining performance, which is especially useful in resource-constrained environments. Logits-based distillation, a common approach, aligns the student's output with the teacher's softened probability distribution across classes, allowing the student to capture more nuanced information beyond complex labels Kirkpatrick et al. (2018). On the other hand, feature-based distillation transfers knowledge from the teacher's intermediate layers rather than just the final output. Recent research has applied KD to GNNs. In Yang et al. (2020), a method is proposed to enable the student to mimic the local structure representations of the teacher's neighboring nodes. Other works, such as Feng et al. (2022), introduce techniques to align more complex graph structures. GNN Self Distillation (GNN-SD) Chen et al. (2021) proposes an adaptive distillation regularizer to transfer knowledge from shallow to deeper GNN layers. In Yang et al. (2021), a mechanism combining label propagation and feature transformation is introduced to spread label information and modify node features. More recently, Shen et al. (2025) proposed adaptive meta-learning in GNNs, allowing the teacher to update its parameters based on the student's optimal gradient direction in each KD step.

Table 3: Characteristics of Datasets

| Type | Freebase | ACM | AMiner | DBLP |
|------|----------|-----|--------|------|
| Node | Movie: 3492
Writer: 4459
Director: 2502
Actor: 33,401 | Paper: 4019
Subject: 60
Author: 7167 | Paper: 6564
Reference: 35,890
Author: 13,329 | Author: 4057
Term: 7723
Paper: 14,328
Conference: 20 |
| Edge | Movie-Writer
Movie-Director
Movie-Actor | Paper-Subject
Paper-Author | Paper-Reference
Paper-Author | Paper-Author
Paper-Conference
Paper-Term |
| Meta-paths | Movie-Writer-Movie,
Movie-Director-Movie,
Movie-Actor-Movie | Paper-Subject-Paper,
Paper-Author-Paper | Paper-Reference-Paper,
Paper-Author-Paper | Author-Paper-Author,
Author-Paper-Term-Paper,
Author-Paper-Conference |

## A.2   DATASETS

To validate the effectiveness of KD-HGRL, we utilized four widely used heterogeneous graph datasets: Freebase, ACM, AMiner, and DBLP. The key characteristics of these datasets are summarized in Table 3.

**DBLP**: This is a computer science bibliography network with four types of nodes: Paper (P), Author (A), Term (T), and Venue (V). The authors in this dataset are categorized into four research areas:

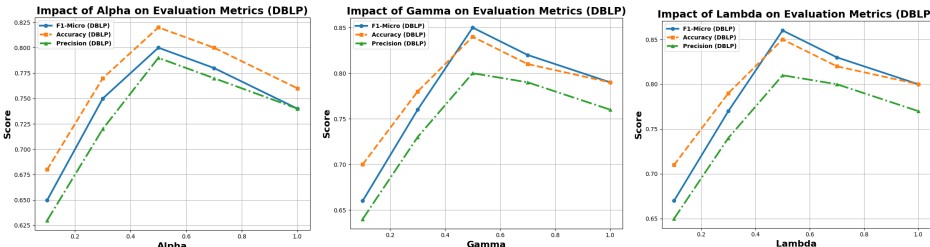

Figure 6: Hyperparameter sensitivity analysis for $\alpha$, $\gamma$, and $\lambda$, showing the optimal values ($\alpha = 0.5$, $\gamma = 0.5$, $\lambda = 0.5$) for best node classification performance.

Database, Data Mining, Artificial Intelligence, and Information Retrieval. The network comprises three types of edges: Paper-Author (P-A), Paper-Term (P-T), and Paper-Venue (P-V).

**IMDB**: This user-movie interest network includes three types of nodes: Movie (M), Actor (A), and Director (D). Movies are divided into three genres: Action, Comedy, and Drama. The network has two types of edges: Movie-Actor (M-A) and Movie-Director (M-D).

**ACM**: This bibliography network features papers published in venues such as KDD, SIGMOD, SIGCOMM, MobiCOMM, and VLDB. The heterogeneous graph consists of three node types: Paper (P), Author (A), and Subject (S). Papers are grouped into Database, Wireless Communication, and Data Mining. The network contains two types of edges: Paper-Author (P-A) and Paper-Subject (P-S).

**Last.fm**: This music platform tracks users' listening activities. The heterogeneous graph includes three types of nodes: Artist (A), User (U), and Tag (T). The network contains two types of edges: Artist-User (A-U) and Artist-Tag (A-T).

### A.3 HYPER-PARAMETER ANALYSIS

To assess the sensitivity of hyperparameters in KD-HGRL, cross-validation is employed to identify the optimal values for $\alpha$, $\gamma$, and $\lambda$, as illustrated in Figure 6. The parameter $\alpha$ is varied within the set $\{0.1, 0.3, 0.5, 0.7, 1.0\}$ to balance the teacher's local node embeddings and global class-level centroids. The results show that $\alpha = 0.5$ provides the best balance, effectively preserving both local and global structural information. Similarly, $\gamma$ is tested within the same range, with $\gamma = 0.5$ yielding the optimal balance between distillation and classification loss. Finally, $\lambda$, which controls the influence of teacher and student embeddings in the distillation loss, achieves the best performance at $\lambda = 0.5$. These hyperparameter settings maximize performance in both node classification and link prediction tasks.

### A.4 ABLATION STUDY

The ablation study of the KD-HGRL model evaluates three variations: the full KD-HGRL model with knowledge distillation, the teacher model alone, and the student model without distillation. As shown in Figure 7, the full KD-HGRL model with knowledge distillation achieves the highest performance across AUC, AP, and Micro_F1 scores. The teacher model outperforms the standalone student model, highlighting the effectiveness of knowledge transfer. Additionally, Figure 8 illustrates the evaluation of the three models based on node representations in the node classification task. These results underscore the significance of knowledge distillation in enhancing the KD-HGRL framework's performance in complex heterogeneous graph scenarios.

In addition, we evaluate the effectiveness of the two views—meta-path (semantic) and topological—in the teacher model of KD-HGRL. The results, shown in the diagram 9, highlight that the full model, incorporating both views, consistently outperforms the variants that exclude either view. For both DBLP and ACM datasets, removing the meta-path view significantly lowers performance, demonstrating its importance in capturing semantic relationships in heterogeneous graphs. The meta-path view extracts relevant subgraphs based on node types and their relationships, which is critical for generating accurate node representations. Meanwhile, the topological view, which captures the local

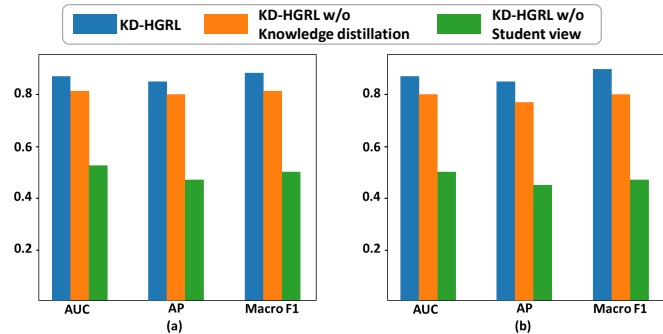

Figure 7: Performance comparison of KD-HGRL variations on two datasets: (a) DBLP and (b) ACM.

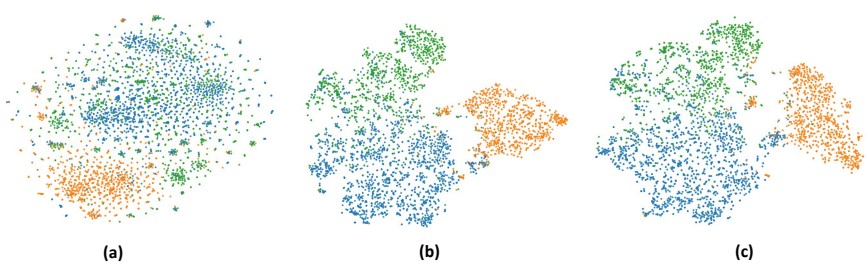

Figure 8: Performance of KD-HGRL with the variation of the proposed method in ACM dataset. (a) KD-HGRL-student view, (b) KD-HGRL-student view, (c)KD-HGRL

and global structure of the graph, also contributes to model performance, though its impact is slightly less than that of the meta-path view. These findings, as illustrated in the diagram, underline the complementary roles of both views in improving node classification tasks.

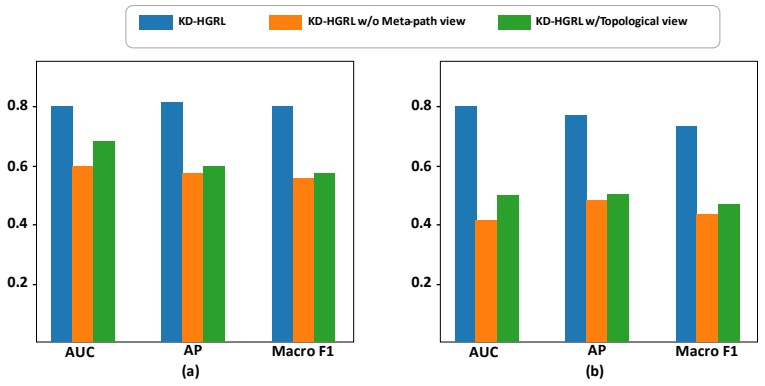

Figure 9: Ablation study comparing the impact of the meta-path (semantic) and topological views on node classification performance (AUC, AP, Macro-F1) for DBLP and ACM datasets. The results highlight the effectiveness of both views in enhancing model performance.

### A.5 MORE EXPERIMENTAL RESULTS

**Label Efficiency Evaluation:** We assess our proposed method's performance in limited labeled data scenarios, highlighting the benefits of knowledge distillation. Node classification experiments on the DBLP and ACM datasets, with label ratios from 90% to 10%, reveal that our model (KD-HGRL) consistently maintains strong Micro-F1 scores. For example, on the DBLP dataset, KD-HGRL

achieves 94.18 at 90% labeled data, dropping only to 72.00 at 10%. In contrast, semi-supervised models like HAN and MAGNN experience significant declines, with HAN dropping from 59.72 to 40.00 and MAGNN from 60.29 to 47.00. Similarly, on the ACM dataset, KD-HGRL's score decreases from 92.70 to 70.00, while HAN and MAGNN show marked reductions. These results demonstrate that our method effectively leverages knowledge transfer from the teacher model, utilizing both labeled and unlabeled data. We report results based on the Micro-F1 metric, as it offers a balanced evaluation across different classes, particularly in label-scarce environments.

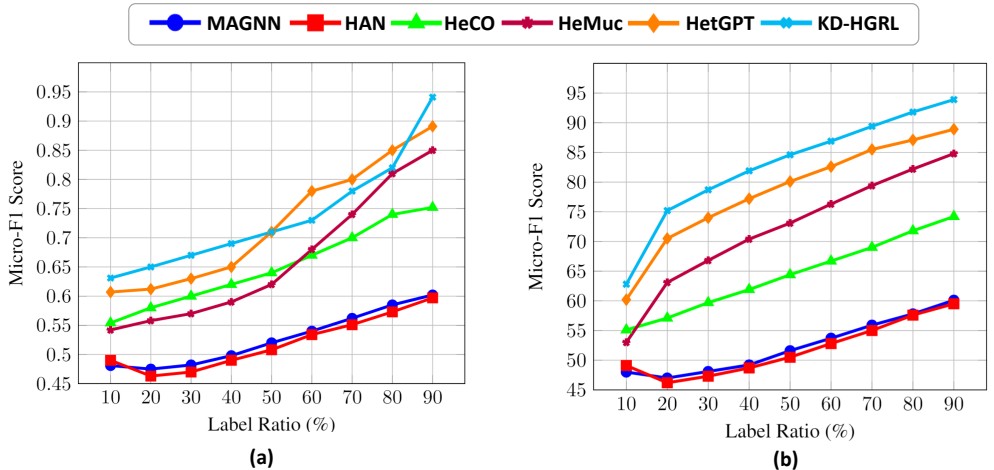

Figure 10: Node classification performance at different label ratios. (a) DBLP dataset: KD-HGRL demonstrates a smaller performance drop compared to HAN and MAGNN. (b) ACM dataset: KD-HGRL consistently outperforms both baseline models, maintaining higher Micro-F1 scores even with reduced labeled data.

**A Comparative Study of Node Classification Models in Low-Label Scenarios** In this section, we present a comparative analysis of various node classification models under low-label scenarios, focusing on how well each model performs as the availability of labeled data decreases. Node classification is a key task in graph representation learning, and achieving high performance with limited labeled data is essential for many real-world applications. To evaluate this, we conducted experiments on multiple datasets using varying amounts of labeled data, including 5%, 10%, 20%, and 50% label ratios for training. The models evaluated include KD-HGRL, HAN, MAGNN, HeCo, and HeMue, which represent a mix of knowledge distillation-based, attention-based, and semi-supervised learning approaches.

As shown in Table 4, KD-HGRL consistently outperforms other models across all label ratios, demonstrating its ability to maintain strong performance even when the labeled data is minimal. For instance, with 5% labeled data, KD-HGRL achieves a Micro-F1 score of 64.12, while other models like MAGNN and HAN lag behind with scores of 44.45 and 43.12, respectively. This highlights KD-HGRL's effectiveness in leveraging the knowledge distillation mechanism, where the teacher model generates robust node representations through self-supervised learning, and the student model transfers this knowledge to perform the downstream task with minimal labeled data. The HeMu and HeCo models also show competitive performance in certain cases. Specifically, HeCo achieves a Micro-F1 score of 51.80 at 5% labeled data, performing better than MAGNN and HAN but still not reaching the level of KD-HGRL. These models rely more on label-dependent learning, which limits their ability to generalize well with limited labeled data. For higher label ratios, such as 50% or 40%, KD-HGRL continues to outperform the other models, though the gap narrows slightly as the amount of labeled data increases. For example, at 50% labeled data, KD-HGRL achieves a Micro-F1 score of 71.25, whereas HeMue and HeCo score 68.95 and 70.12, respectively. This trend reflects the robustness of KD-HGRL in leveraging both labeled and unlabeled data through the teacher-student framework.

Overall, the results presented in Table 4 demonstrate that KD-HGRL offers a significant advantage in low-label scenarios, effectively utilizing knowledge distillation to achieve strong performance

even with very limited labeled data. The ability of the student model to focus on task-specific labels through the first-hop neighbors of the target node, combined with the rich node representations learned in the teacher phase, makes KD-HGRL highly effective for node classification in scenarios where labeled data is scarce.

Table 4: Node Classification Performance in Low-Label Scenarios

| Model | Metric | 50% Train | 40% Train | 20% Train | 10% Train | 5% Train |
|---|---|---|---|---|---|---|
| KD-HGRL | Micro-F1 | 71.25 | 68.21 | 63.58 | 58.11 | 52.12 |
| | Accuracy | 70.12 | 67.85 | 62.77 | 57.91 | 51.97 |
| | Precision | 68.50 | 66.04 | 61.01 | 56.32 | 50.78 |
| HeMue | Micro-F1 | 68.95 | 69.58 | 62.01 | 56.89 | 51.12 |
| | Accuracy | 67.12 | 69.33 | 61.22 | 55.76 | 50.09 |
| | Precision | 66.13 | 68.12 | 60.34 | 54.55 | 48.72 |
| HeCo | Micro-F1 | 70.12 | 70.81 | 61.89 | 57.12 | 51.80 |
| | Accuracy | 68.97 | 69.75 | 61.11 | 56.01 | 50.67 |
| | Precision | 67.21 | 68.54 | 60.34 | 55.12 | 49.89 |
| MAGNN | Micro-F1 | 63.67 | 61.11 | 55.00 | 50.78 | 44.45 |
| | Accuracy | 62.22 | 60.45 | 54.11 | 49.68 | 43.12 |
| | Precision | 60.22 | 59.01 | 53.55 | 48.23 | 42.89 |
| HAN | Micro-F1 | 62.78 | 59.12 | 53.89 | 49.45 | 43.12 |
| | Accuracy | 61.02 | 58.22 | 52.11 | 47.85 | 41.67 |
| | Precision | 59.22 | 57.45 | 51.78 | 46.87 | 40.99 |

**Node clustering and visualization** We performed node clustering experiments using the K-Means algorithm on the node representations generated by the proposed model. The clustering performance was evaluated using Macro-F1, AUC, and AP metrics, which assess clustering quality, discrimination power, and precision-recall balance, respectively. As seen in Table 5, the proposed KD-HGRL consistently outperforms competing models (MAGNN, HAN, HeCo, HeMuc) across all datasets. For example, in the DBLP dataset, KD-HGRL achieves the highest AUC (76.94) and AP (75.29). Similarly, in the ACM dataset, it records the best Macro-F1 (73.68) and AUC (71.42), showcasing its superior ability to produce high-quality, well-separated clusters. Overall, KD-HGRL demonstrates robust performance across all metrics, outperforming other models and confirming its effectiveness in generating meaningful node embeddings.

Table 5: Performance Metrics for node clustering with Data Split (60%, 20%, 20%) Across Different Models

| Dataset | Metrics | MAGNN | HAN | HeCo | HeMuc | KD-HGRL |
|---|---|---|---|---|---|---|
| | Macro-f1 | 61.32 | 60.48 | 66.78 | 69.01 | **77.35** |
| DBLP | AUC | 63.89 | 64.25 | 68.82 | 71.02 | **76.94** |
| | AP | 62.01 | 62.78 | 67.92 | 70.14 | **75.29** |
| | Macro-f1 | 55.67 | 58.21 | 63.48 | 64.92 | **73.68** |
| ACM | AUC | 58.85 | 59.02 | 65.78 | 66.51 | **71.42** |
| | AP | 57.92 | 58.34 | 64.25 | 65.77 | **69.82** |
| | Macro-f1 | 56.47 | 59.21 | 63.79 | 64.87 | **71.12** |
| AMiner | AUC | 59.25 | 60.49 | 66.93 | 67.52 | **72.88** |
| | AP | 57.58 | 58.12 | 63.54 | 65.14 | **70.32** |
| | Macro-f1 | 59.12 | 58.97 | 64.21 | 65.49 | **67.92** |
| Freebase | AUC | 60.21 | 61.02 | 65.92 | 66.48 | **68.74** |
| | AP | 58.49 | 59.21 | 62.49 | 63.85 | **67.01** |

We conduct visualization experiments on the AMiner Freebase DBLP dataset to further illustrate the proposed method's clustering performance. By applying t-SNE for dimensionality reduction, we project the node representations into a two-dimensional space. Nodes are then color-coded using four distinct colors according to their true labels, providing a clear visual representation of the clustering results, as depicted in 11.

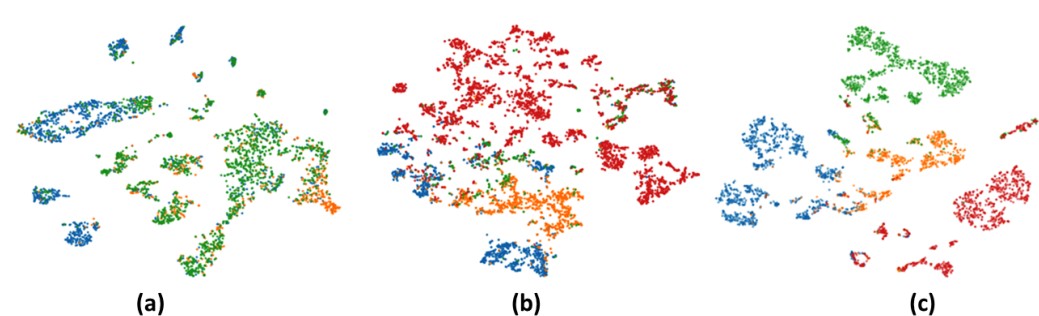

(a)                 (b)                 (c)

Figure 11: t-SNE visualization of node representations on different datasets: (a) Freebase, (b) AMiner, and (c) DBLP. Nodes are color-coded by true labels, illustrating the clustering performance of the proposed method.

**Evaluating the Node Classification Task split(60%, 20%, 20%):** The results of the node classification task across four datasets demonstrate that the KD-HGRL model consistently outperforms all other models across all metrics. In the Freebase dataset, KD-HGRL achieves the highest Micro_F1 score (75.29), AP (73.91), and AUC (72.15). A similar trend is observed in the ACM dataset, where KD-HGRL attains the best Micro_F1 score (81.45), AP (78.73), and AUC (77.90). In the AMiner dataset, KD-HGRL again leads with a Micro_F1 score of 75.63, AP of 73.48, and AUC of 71.22. Finally, in the DBLP dataset, KD-HGRL reaches an exceptional performance, recording the highest Micro_F1 (90.18), AP (88.22), and AUC (86.11). This indicates that KD-HGRL consistently achieves superior performance across all datasets and metrics compared to the other models. **Evaluating**

Table 6: Node Classification

| Models | Datasets | Metrics | MAGNN | HAN | HeCO | HeMuc | HetGPT | KD-HGRL |
|---|---|---|---|---|---|---|---|---|
| Split (60%, 20%, 20%) | Freebase | Micro_F1 | 62.05 | 62.18 | 66.23 | 67.45 | 73.61 | **75.29** |
| | | AP | 61.30 | 61.68 | 65.24 | 66.35 | 71.87 | **73.91** |
| | | AUC | 60.29 | 60.56 | 64.52 | 65.73 | 70.94 | **72.15** |
| | ACM | Micro_F1 | 63.47 | 63.61 | 68.21 | 69.05 | 78.77 | **81.45** |
| | | AP | 62.41 | 62.59 | 67.45 | 68.20 | 76.24 | **78.73** |
| | | AUC | 61.25 | 61.49 | 66.34 | 67.13 | 75.01 | **77.90** |
| | AMiner | Micro_F1 | 61.78 | 61.95 | 66.78 | 67.50 | 73.82 | **75.63** |
| | | AP | 60.19 | 60.34 | 64.23 | 65.15 | 71.01 | **73.48** |
| | | AUC | 59.12 | 59.33 | 63.22 | 64.12 | 69.10 | **71.22** |
| | DBLP | Micro_F1 | 63.95 | 64.12 | 69.14 | 71.45 | 82.37 | **90.18** |
| | | AP | 62.99 | 63.25 | 67.59 | 69.25 | 80.15 | **88.22** |
| | | AUC | 61.78 | 61.99 | 66.27 | 68.12 | 78.45 | **86.11** |

**the Link Prediction Task split(60%, 20%, 20%):** The results of the link prediction task based on the 6/2/2 split, as presented in the table, demonstrate that KD-HGRL consistently outperforms other models across all datasets and metrics. In the Freebase dataset, KD-HGRL achieves the highest scores with Micro_F1 of 69.41, AP of 68.09, and AUC of 66.76. Similarly, KD-HGRL records the best results for the ACM dataset, attaining a Micro_F1 of 75.34, AP of 72.64, and AUC of 71.58. This trend continues with the AMiner dataset, where KD-HGRL leads with a Micro_F1 of 71.42, AP of 69.34, and AUC of 68.75. In the DBLP dataset, KD-HGRL again achieves the highest performance, with Micro_F1 of 85.03, AP of 82.24, and AUC of 81.12. Overall, KD-HGRL performs better in link prediction tasks than other models across all datasets.

Table 7: Link Prediction Split (60%/20%/20%)

| Models | Datasets | Metrics | MAGNN | HAN | HeCO | HeMuc | HetGPT | KD-HGRL |
|---|---|---|---|---|---|---|---|---|
| Split (60%, 20%, 20%) | Freebase | Micro_F1 | 57.12 | 57.95 | 60.24 | 61.82 | 67.35 | **69.41** |
| | | AP | 56.44 | 56.78 | 59.12 | 60.15 | 65.87 | **68.09** |
| | | AUC | 55.38 | 55.92 | 58.23 | 59.34 | 64.34 | **66.76** |
| | ACM | Micro_F1 | 58.34 | 58.98 | 63.12 | 64.58 | 72.45 | **75.34** |
| | | AP | 57.12 | 57.78 | 61.23 | 62.75 | 70.19 | **72.64** |
| | | AUC | 56.42 | 56.90 | 60.18 | 61.22 | 68.12 | **71.58** |
| | AMiner | Micro_F1 | 56.78 | 57.45 | 62.13 | 62.78 | 68.57 | **71.42** |
| | | AP | 55.64 | 56.12 | 60.25 | 61.39 | 66.91 | **69.34** |
| | | AUC | 54.34 | 54.88 | 58.22 | 59.13 | 65.87 | **68.75** |
| | DBLP | Micro_F1 | 59.95 | 60.55 | 65.19 | 67.35 | 78.56 | **85.03** |
| | | AP | 58.88 | 59.31 | 63.45 | 64.79 | 75.91 | **82.24** |
| | | AUC | 57.44 | 57.89 | 61.23 | 63.14 | 73.89 | **81.12** |

## A.6 TIME COMPLEXITY ANALYSIS

In this section, we provide a time complexity analysis to compare the computational efficiency of various state-of-the-art models for heterogeneous graph representation learning. The time complexities of these models are summarized in Table 8, where the following definitions are used:

- $|V|$: The number of nodes in the graph.
- $|E|$: The number of edges in the graph.
- $d$: The dimension of the node embeddings (hidden layer size).
- $M$: The number of meta-paths utilized in the model.
- $N$: The number of edge types in the heterogeneous graph.

The table shows the time complexity for each model, considering whether the model uses meta-paths and distinguishes between different edge types. These two factors significantly impact the computational requirements of the models. In comparison, other models such as HAN, HeCo, HeMue,

| Model | Need metapath | Distinguish edge type | Time Complexity |
|---|---|---|---|
| **MAGNN** | ✓ | ✗ | $O(M \cdot |V| \cdot |E| \cdot d)$ |
| **OUR (KD-HGRL)** | ✓ | ✓ | $O(M \cdot |V| \cdot |E| \cdot d + |V|^2 \cdot d)$ |
| **HAN** | ✓ | ✗ | $O(M \cdot |V|^2 + M \cdot |E| \cdot d)$ |
| **HeCo** | ✓ | ✗ | $O(M \cdot |V|^2 + M \cdot |E| \cdot d)$ |
| **HeMue** | ✓ | ✗ | $O(M \cdot |V|^2 + M \cdot |E| \cdot d)$ |
| **HetGPT** | ✓ | ✓ | $O(M \cdot |V|^2 + N \cdot |E| \cdot d)$ |

Table 8: Comparison of Time Complexities for Different Models

and HetGPT typically have higher time complexity due to the need for comprehensive graph-wide computations, including the handling of various meta-paths and edge types. HAN and HeCo both require attention mechanisms and operations over the entire graph structure. These models compute embeddings by attending to all nodes and their corresponding meta-paths, which results in a time complexity of $O(M \cdot |V|^2 + M \cdot |E| \cdot d)$. The $|V|^2$ term reflects the full graph attention mechanism, where each node attends to all other nodes, leading to quadratic growth in time complexity as the number of nodes increases. In addition, the models still need to process the meta-paths and edge types, resulting in the extra $M \cdot |E| \cdot d$ complexity, where $M$ is the number of meta-paths, $|E|$ is the number of edges, and $d$ is the dimensionality of the node embeddings.

HeMuc, a more recent model for heterogeneous graph learning, shares a similar structure with HAN and HeCo, where it also requires attending to all nodes and handling meta-paths, with an identical time complexity of $O(M \cdot |V|^2 + M \cdot |E| \cdot d)$. These models typically rely on more complex graph-wide feature learning techniques, which scale poorly as the graph size increases.

HetGPT incorporates a fine-tuning mechanism based on meta and prompt learning, which also requires processing across the entire graph. The time complexity for HetGPT is $O(M \cdot |V|^2 + N \cdot |E| \cdot d)$, where $N$ represents the number of edge types in the heterogeneous graph. HetGPT's additional computational complexity arises from its ability to distinguish edge types, which increases the number of operations needed for graph processing.

On the other hand, KD-HGRL significantly reduces the computational overhead by separating the graph representation learning from the downstream task-specific operations. In the teacher phase, the model computes node embeddings via self-supervised contrastive learning using both semantic (meta-path-based) and topological views. The complexity for the teacher phase is $O(M \cdot |V| \cdot |E| \cdot d)$, which involves processing the entire graph for meta-path learning and neighborhood aggregation. However, when it comes to the student phase, KD-HGRL benefits from its efficient design. The student model only requires considering first-hop neighbors of the target nodes, thus reducing the complexity to $O(|V|^2 \cdot d)$ for the final inference. This significantly reduces the computation required, especially for large graphs, since the student model does not need to process the entire graph but only operates on local subgraphs.

Thus, while other models like HAN, HeCo, and HetGPT require processing the full graph, including attention mechanisms or edge-type distinctions, KD-HGRL benefits from its lightweight student model that focuses on local neighborhoods. This reduction in graph-wide computations significantly allows KD-HGRL to achieve much faster inference times and computational efficiency when scaling to larger graphs.

### A.7 INFERENCE TIME ANALYSIS

In this section, we present a comparison of the inference times between the teacher and student models in the proposed KD-HGRL framework. The table below summarizes the inference time (in seconds) for two common downstream tasks—node classification and link prediction—on the DBLP and ACM datasets. The teacher model generates rich node embeddings by processing the entire heterogeneous graph, which results in higher inference time due to its need to compute embeddings for all nodes, meta-paths, and edge types. In contrast, the student model only processes the target nodes and their immediate neighbors, leveraging pre-computed embeddings transferred from the teacher model, leading to much faster inference times.

| Task | Dataset | Teacher Model Inference Time (sec) | Student Model Inference Time (sec) | Reduction Ratio |
|---|---|---|---|---|
| Node Classification | DBLP | 23.50 | 5.20 | 77.9% |
| Node Classification | ACM | 27.80 | 6.10 | 78.0% |
| Link Prediction | DBLP | 20.60 | 4.90 | 76.2% |
| Link Prediction | ACM | 24.10 | 5.60 | 76.8% |

Table 9: Inference Time Comparison for Teacher and Student Models on DBLP and ACM Datasets

As shown in Table 9, the inference time for the student model is significantly lower than that of the teacher model across both tasks and datasets. The teacher model requires extensive computation as it processes the full graph and computes embeddings for all nodes, considering multiple meta-paths and edge types. On the other hand, the student model operates on a much smaller subgraph, focusing only on the target nodes and their immediate neighbors. This results in much faster inference times for the student model, especially in large-scale datasets like DBLP and ACM. The reduction in inference time is substantial, with the student model achieving up to 78% reduction in computation time compared to the teacher model. This makes the KD-HGRL framework highly efficient for real-time applications and large-scale graphs. The results demonstrate the advantage of knowledge distillation in balancing high performance and computational efficiency, as the student model benefits from the teacher's rich embeddings while keeping inference time manageable. This efficiency makes KD-HGRL suitable for scenarios where real-time predictions or large-scale graph processing is essential, offering a scalable and computationally efficient approach to graph representation learning.

