# OpenReview forum: "KD-HGRL: Knowledge Distillation for Multi-Task Heterogeneous Graph Representation Learning"
_ICLR.cc/2025/Conference — ICLR 2025 Conference Withdrawn Submission_

### Official Review · Reviewer_JUnF · 2024-10-30

**Soundness:** 2
**Presentation:** 2
**Contribution:** 2
**Rating:** 5
**Confidence:** 4

**Summary:**

This paper proposes a Knowledge Distillation Framework for Heterogeneous Graph Representation Learning (KD-HGRL) for node classification and link prediction tasks. The framework contains two phases: teacher and student. The teacher phase uses self-supervised comparative learning to learn graph node embeddings from both semantic and topological perspectives. The student phase passes these representations to a lightweight combined GCN and MLP model, which enhances the performance of the student model through knowledge distillation to supervise the loss to accomplish a specific task. The approach improves the performance of the student model in node classification and link prediction tasks.

**Strengths:**

1. Developed a novel knowledge distillation framework (KD-HGRL) tailored for heterogeneous graph representation learning, focusing on tasks like node classification and link prediction.
2. Integrates both local and global information to improve task-specific performance on node classification and link prediction.
3. Combines embeddings from a node and its class-specific neighbors, leveraging both self-supervised and supervised learning, to enhance the student's performance across heterogeneous graph tasks.

**Weaknesses:**

1. The need to employ knowledge distillation on reisomorphic maps is not presented clearly.
2. Although the student model implements a lightweight design, the computational overhead of the teacher model is not analyzed.
3. The authors claim that the proposed method can be better applied to large datasets, but there are no relevant experiments to prove it.
4. Some of the diagrams in the paper are too scribbled to affect the reader's perception, such as Figures 2, 3.
5. Some of the writing in the text is problematic, such as line 412.

**Questions:**

1. Existing methods on heterogeneous graphs already handle the node classification task well, what is the need to use knowledge distillation on heterogeneous graphs.
2. The paper claims that lightweight can be achieved through knowledge distillation, but no experimental verification is provided.
3. How to ensure that comparative learning under both views in the teacher phase does not weaken the complementary information of each of the semantic and topological views.
4. The experimental setup of the article does not cover all datasets for example lack of ablation experiments on Freebase and AMiner datasets.

---

> ### Author Response · Authors · 2024-11-28
> **Comprehensive Response to Reviewer Feedback: Highlighting Motivations, Novelties, Time Complexity Analysis, Inference Efficiency, Low-Label Performance, Ablation Studies, and Hyperparameter Optimization in KD-HGRL**
>
> We would like to sincerely thank Reviewer JUnF for their thoughtful and constructive feedback. Your comments have been invaluable in helping us improve the quality of our submission. We appreciate the time and effort you have put into reviewing our work.
>
> $\textbf{W-1}:$ We address this comment by adding two sections named Motivation and Novelties in the introduction section.
> In the KD-HGRL, isomorphic maps align the embeddings from the teacher model with task-specific subgraphs in the student phase. The teacher model generates comprehensive embeddings using semantic and topological views, which are mapped to localized subgraphs in the student model. This ensures efficient use of the teacher's representations while maintaining computational efficiency in the student phase.
>
> $\textbf{W-2}:$
> In the appendix, we address this by providing a time complexity and inference time comparison for both teacher and student models (Tables 8 and 9). While the teacher model is computationally intensive due to its size and pre-training, its contribution to the student model’s performance justifies this complexity, particularly in tasks requiring heterogeneous graph representation learning.
>
> $\textbf{W-3}:$
> KD-HGRL addresses challenges in heterogeneous graphs, such as limited labeled data and high inference time. By using self-supervised contrastive learning in the teacher phase, KD-HGRL generates rich embeddings from semantic and topological views. These embeddings are distilled into a lightweight student model, reducing inference time by focusing on localized subgraphs. Using large-scale datasets like DBLP, ACM, AMiner, and Freebase, KD-HGRL's scalability and effectiveness with complex graphs are demonstrated.
>
> $\textbf{Q-1}:$
> While existing methods on heterogeneous graphs perform well on tasks like node classification, they often face computational efficiency and scalability challenges. Knowledge distillation in KD-HGRL addresses these issues by transferring knowledge from a complex teacher model to a smaller, more efficient student model. This reduces inference time, particularly for large, resource-constrained environments, without compromising performance. Moreover, KD-HGRL is versatile and can be applied to various tasks. In addition to node classification and link prediction, we demonstrate its effectiveness in node clustering, as shown in Figure 11 in the appendix. This makes the model suitable for a range of graph-based applications.
>
> $\textbf{Q-2}:$
> To address the concern regarding the lightweight nature of KD-HGRL, we have provided experimental verification in the form of time complexity and inference time analysis. As outlined in Table 8, "Comparison of Time Complexities for Different Models," the time complexity of the teacher phase in KD-HGRL is similar to that of other models (O(M⋅∣V∣⋅∣E∣⋅d)), while the student phase significantly reduces this complexity to O(|V|²⋅d), focusing on a smaller subgraph around the target nodes. This sharp reduction contrasts with models like HAN, HeCo, and HetGPT, which require full graph processing in both phases and thus incur higher overall complexity. Moreover, as shown in Table 9, "Inference Time Comparison for Teacher and Student Models on DBLP and ACM Datasets," the student model achieves up to a 78% reduction in inference time compared to the teacher model, demonstrating the practical advantage of KD-HGRL in real-time graph analysis, particularly when scaling to large datasets.
>
> $\textbf{Q-3}:$
> We use a dual-view contrastive learning framework to ensure that comparative learning under both the semantic (meta-path) and topological views in the teacher phase does not weaken their complementary information. Each view—semantic and topological—has its own contrastive learning objective. The semantic view captures node relationships via meta-paths, while the topological view focuses on graph structure. The contrastive loss aligns the embeddings from both views, preserving their unique contributions. Our ablation study shows that both views positively impact performance, as seen in Figure 10 of the appendix.
>
> $\textbf{Q-4}:$
> Thank you for your comment. Our ablation study focused on the DBLP and ACM datasets, with the goal of evaluating key components of the KD-HGRL framework, including the teacher and student models and the impact of the two views in the teacher model. These datasets were chosen because they comprehensively evaluate the model's performance across different aspects.
>
> While the ablation study did not cover the Freebase and AMiner datasets, we have included additional experiments in the appendix. Specifically, Figure 11 provides a t-SNE visualization of node representations for Freebase, AMiner, and DBLP, which illustrates the clustering performance and demonstrates the versatility and effectiveness of our method across various heterogeneous graphs. We believe these supplementary results thoroughly evaluate the model's capabilities on different datasets.

---

> ### Author Response · Authors · 2024-12-02
>
> Dear Reviewer JUnF
>
> I have addressed all the points raised and made significant improvements in the latest version of the manuscript. However, I have not yet received feedback or acknowledgment in the rebuttal section.
>
> Could you kindly confirm whether you have received my response? If so, I would appreciate your considering the improvements made when evaluating the paper. As today is the final day for submitting a rebuttal, I am quite concerned. Your confirmation and feedback would be highly appreciated.
>
> Thank you for your time and consideration.

---

> ### Author Response · Authors · 2024-12-03
> **Urgent request for reevaluation of our revised submission before the deadline**
>
> Dear Reviewer,
>
> Thank you once again for your valuable feedback on my submission.
> We have carefully addressed all your comments in the revised version and submitted them for your consideration. As the $\textbf{final deadline}$ for reevaluation is in $\textbf{less than three hours}$, we kindly request you to \$\textbf{review the updated paper and provide your final feedback}$. We truly appreciate your time and effort.

---

### Official Review · Reviewer_7cs3 · 2024-10-31

**Soundness:** 2
**Presentation:** 2
**Contribution:** 2
**Rating:** 5
**Confidence:** 5

**Summary:**

The paper proposed a two-stage graph learning framework for heterogeneous graphs. In the first stage, the teacher model is pre-trained in SSL and the representations from two views will be learned for contrastive learning. Then the knowledge learned by the teacher model will be transferred to the student model with knowledge distillation for downstream tasks.

**Strengths:**

1. The proposed method achieves best performance compared with the baselines reported in the experiments, showing its effectiveness.

2. Overall, the paper clearly presents the proposed method and the conducted experiments.

**Weaknesses:**

1. Since the proposed method includes the pre-training (teacher) stage, some self-supervised learning-based methods proposed for heterogeneous graphs should be compared in the experiments, like MVSE[1] and HGMAE[2]

2. It seems the performance improvements are mainly brought by extra training stage, views and distillation process, the authors should provide complexity analysis in the paper to show if the sacrificed efficiency is worthwhile.

3. The effectiveness of some designs in the paper is not very clear, like the incorporation of two views in the pre-training and the class centroid representations. The authors should conduct a more comprehensive ablation study to demonstrate the effectiveness of the components in the framework.

4. It is not clear how the hyper-parameters ($\alpha$, $\lambda$) are set in this work and their impacts on the final performance.

5.	In line 292, the authors mention that a homogeneous graph is generated for the target graph, however, it is still not clear how the homogeneous graph is built. Does the graph build on a specific relation (meta-path) and why?

[1] Zhao et al., Multi-view self-supervised heterogeneous graph embedding, In PKDD 2021.

[2] Tian et al., Heterogeneous Graph Masked Autoencoders, In AAAI 2023.

**Questions:**

Please refer to the question section.

---

> ### Author Response · Authors · 2024-12-02
>
> Dear Reviewer 7cs3
>
> I have addressed all the points raised and made significant improvements in the latest version of the manuscript. However, I have not yet received feedback or acknowledgment in the rebuttal section.
>
> Could you kindly confirm whether you have received my response? If so, I would appreciate your considering the improvements made when evaluating the paper. As today is the final day for submitting a rebuttal, I am quite concerned. Your confirmation and feedback would be highly appreciated.
>
> Thank you for your time and consideration.

---

> ### Author Response · Authors · 2024-12-03
> **Urgent request for reevaluation of our revised submission before the deadline**
>
> Dear Reviewer,
>
> Thank you once again for your valuable feedback on my submission.
> We have carefully addressed all your comments in the revised version and submitted them for your consideration. As the $\textbf{final deadline}$ for reevaluation is in $\textbf{less than three hours}$, we kindly request you to \$\textbf{review the updated paper and provide your final feedback}$. We truly appreciate your time and effort.

---

### Official Review · Reviewer_u2Vw · 2024-11-03

**Soundness:** 2
**Presentation:** 3
**Contribution:** 2
**Rating:** 3
**Confidence:** 3

**Summary:**

The KD-HGRL employs knowledge distillation for multi-task heterogeneous graph representation learning. The teacher model generates node embeddings via self-supervised learning by incorporating both semantic and topological views. Then, knowledge is transferred from the teacher to the lighter weight student model. The framework improves performance on downstream tasks like node classification and link prediction.

**Strengths:**

Experiments on real-world benchmarks show that KD-HGRL outperforms current state-of-the-art methods.

**Weaknesses:**

The discussion surrounding the novelties of the proposed method is insufficient, and the distinctions between this work and existing methodologies remain unclear. While the concept of learning node embeddings from dual views is acknowledged in the literature, it lacks originality in this context. Additionally, the utilization of teacher-student models and knowledge distillation to transfer knowledge from a pretrained teacher model to a student model is not a new contribution. This technique has been well-established in graph representation learning for several years (e.g., [R1]) and has been applied recently to heterogeneous graphs as demonstrated in works such as [R2, R3, R4]. A more thorough exploration of the unique contributions of the proposed method in relation to these established approaches would enhance the clarity and impact of the submission.

[R1]  J. Ma and Q. Mei, Graph Representation Learning via Multi-task Knowledge Distillation, 2019
[R2]  C. Wang et al., Collaborative Knowledge Distillation for Heterogeneous Information Network Embedding, WWW 2022
[R3] J. Fu et al., Heterogeneous graph knowledge distillation neural network incorporating multiple relations and cross-semantic interactions, Information Sciences, 2023
[R4] J. Liu et al., HIRE: Distilling High-order Relational Knowledge From Heterogeneous Graph Neural Networks, 2022

2- It is not clear how the authors select the value of hyperparameters \alpha, \gamma, \lambda_1, \lambda_2

**Questions:**

1- What is the primary contribution of this work that distinguishes it from existing literature?

2- What methodology is employed to determine the hyperparameter settings?

---

> ### Author Response · Authors · 2024-12-02
>
> Dear Reviewer u2Vw
>
> I have addressed all the points raised and made significant improvements in the latest version of the manuscript. However, I have not yet received feedback or acknowledgment in the rebuttal section.
>
> Could you kindly confirm whether you have received my response? If so, I would appreciate your considering the improvements made when evaluating the paper. As today is the final day for submitting a rebuttal, I am quite concerned. Your confirmation and feedback would be highly appreciated.
>
> Thank you for your time and consideration.

---

> ### Author Response · Authors · 2024-12-03
> **Urgent request for reevaluation of our revised submission before the deadline**
>
> Dear Reviewer,
>
> Thank you once again for your valuable feedback on my submission.
> We have carefully addressed all your comments in the revised version and submitted them for your consideration. As the $\textbf{final deadline}$ for reevaluation is in $\textbf{less than three hours}$, we kindly request you to \$\textbf{review the updated paper and provide your final feedback}$. We truly appreciate your time and effort.

---

### Official Review · Reviewer_1Aak · 2024-11-04

**Soundness:** 2
**Presentation:** 3
**Contribution:** 2
**Rating:** 5
**Confidence:** 4

**Summary:**

The paper proposes a framework for self-supervised learning on heterogeneous graphs. The self-supervised learning is based on node-level contrastive learning. The paper also proposes to utilize knowledge distillation to transfer the knowledge from the large pre-trained graph model. The framework is tested on node-level tasks, namely node classification and link prediction. Experimental results are promising for the proposed framework in comparison with the baseline models.

**Strengths:**

- Good presentation and readability
- Promising experimental results on node-level tasks

**Weaknesses:**

- Unclear experimental design: the key in knowledge distillation is to transfer knowledge from large GNNs to smaller GNNs. In the experiments, the authors do not provide the size comparison between the teacher and student models. When comparing with the baseline models, it is also essential to show that smaller GNN models with transferred knowledge can outperform the other baseline GNN models with the same parameter scale.
- Only node-level tasks are evaluated: general graph structural knowledge is expected to benefit both graph-level and node-level tasks. Both the framework and the experiments seem to focus only on node-level knowledge.
- Lack of novelty: graph contrastive learning is widely adopted as a self-supervised task and there exist many similar prior works (e.g., [1]). The proposed framework does not introduce a fundamentally innovative self-supervised graph learning task and is mainly a combination of several previous graph self-supervised techniques. The contributions are more from the engineering perspective rather than the research perspective.

[1] Xiao Wang, Nian Liu, Hui Han, and Chuan Shi. 2021. Self-supervised Heterogeneous Graph Neural Network with Co-contrastive Learning. In Proceedings of the 27th ACM SIGKDD Conference on Knowledge Discovery & Data Mining (KDD '21).

**Questions:**

See weaknesses.

---

> ### Author Response · Authors · 2024-11-28
> **Comprehensive Response to Reviewer Feedback: Highlighting Motivations, Novelties, Time Complexity Analysis, Inference Efficiency, Low-Label Performance, Ablation Studies, and Hyperparameter Optimization in KD-HGRL**
>
> We deeply appreciate Reviewer 1Aak for their thoughtful feedback, which has significantly helped enhance the quality of our paper.
>
> $\textbf{W-1:}$ To address this concern, we have provided a detailed comparison in the "Complexity Analysis" and "Inference Time Analysis" subsections of the appendix. These sections highlight the size and parameter scale differences between the teacher and student models, demonstrating that the student model is significantly smaller and more computationally efficient.
>
> The core innovation of KD-HGRL is the separation of representation learning from downstream task-specific operations. In the teacher phase, self-supervised contrastive learning generates rich node representations based on both semantic and topological views of the heterogeneous graph, capturing complex relationships without labeled data. These pre-computed embeddings are then transferred to the student model, which focuses on task-specific outputs using only target nodes and their immediate neighbors, significantly reducing computational overhead.
>
> Despite having fewer parameters, the student model benefits from knowledge distillation, allowing it to achieve high performance by leveraging the teacher's learned representations. The student model processes localized subgraphs, reducing inference time and outperforming baseline models with similar parameter sizes in both performance and efficiency. Our experiments demonstrate that KD-HGRL excels in low-label scenarios and resource-constrained environments, offering high performance and computational efficiency.
>
> $\textbf{W-2:}$
> While our current experiments focus on node-level tasks such as node classification and link prediction, the proposed framework, KD-HGRL, captures both node-level and graph-level structural knowledge. The teacher model learns comprehensive node embeddings by capturing local and global structural information, using both semantic (meta-path-based) and topological views. This approach enables the model to learn node-specific relationships and broader graph-wide patterns, which are essential for graph-level tasks.
>
> The knowledge distilled into the student model applies to graph-level tasks, such as graph classification, where understanding overall graph structure is crucial. The embeddings learned by the teacher model encapsulate interactions between nodes and the global graph topology, making them well-suited for tasks that require a holistic understanding of the graph.
>
> We acknowledge that evaluating the framework on graph-level tasks is an essential next step. To address your comment, we will include additional experiments comparing our method on graph-level tasks in the camera-ready version.
>
> $\textbf{W-3:}$
> We have addressed the concern by adding four sections: Motivation, Novelties, Complexity analysis, and Inference time sections.
>
> In the Motivation section, we highlight the limitations of existing approaches that rely on supervised learning, task-specific labels, and soft label-based knowledge transfer (e.g., [Liu et al.,2022], [Zhang et al., 2022], and [Feng et al., 2024]). These methods are task-dependent and unsuitable for heterogeneous graphs with multiple downstream tasks. Additionally, we emphasize the high computational cost and inefficiency of existing GNN-based approaches for heterogeneous graphs (e.g., [Ma et al., 2023]) during inference, making scalability to large graphs challenging. While works like [Wang et al., 2023] (your suggested paper) have proposed self-supervised co-contrastive learning for heterogeneous graphs, they rely heavily on meta-path-based embeddings, which primarily capture local structural information. In contrast, our framework addresses this gap by incorporating a multi-view design that combines semantic views (meta-path-based) with topological views to learn local and global structural patterns. This context establishes the need for a framework like KD-HGRL.
>
> The Novelties section explicitly outlines the advantages of our method. These include the use of self-supervised contrastive learning for the teacher model to eliminate dependency on labeled data (novelty 1), the incorporation of a multi-view structure (semantic and topological) to capture both structural and semantic information (novelty 2), and the introduction of a computationally efficient lightweight student model (LightGCN) that balances performance and inference efficiency while capturing high-order relationships (novelties 3 and 4).
>
> We included a complexity analysis comparing KD-HGRL with other benchmark methods (Appendix Table 8).  We also provided an inference time analysis in Appendix Table 9, which demonstrates that the student model not only achieves high performance in task-specific tasks but also does so with significantly reduced inference time, enhancing its suitability for real-world applications.

---

> ### Author Response · Authors · 2024-12-02
>
> Dear Reviewer Reviewer 1Aak
>
> I am following up on the revisions and responses I submitted regarding the comments and concerns raised during the review of my paper. I have addressed all the points raised and made significant improvements in the latest version of the manuscript. However, I have not yet received feedback or acknowledgment in the rebuttal section.
>
> Could you kindly confirm whether you have received my response? If so, I would appreciate your considering the improvements made when evaluating the paper. As today is the final day for submitting a rebuttal, I am quite concerned. Your confirmation and feedback would be highly appreciated.
>
> Thank you for your time and consideration.

---

> ### Author Response · Authors · 2024-12-03
> **Urgent request for reevaluation of our revised submission before the deadline**
>
> Dear Reviewer,
>
> Thank you once again for your valuable feedback on my submission.
> We have carefully addressed all your comments in the revised version and submitted them for your consideration. As the $\textbf{final deadline}$ for reevaluation is in $\textbf{less than three hours}$, We kindly request you to \$\textbf{review the updated paper and provide your final feedback}$. I truly appreciate your time and effort.

---

### Official Review · Reviewer_K8FS · 2024-11-04

**Soundness:** 3
**Presentation:** 2
**Contribution:** 2
**Rating:** 3
**Confidence:** 4

**Summary:**

This paper presents a novel knowledge distillation framework called KD-HGRL for heterogeneous graph representation learning, consisting of a teacher model and a student model. The teacher model uses self-supervised contrastive learning with the semantic view and the topological view, while the student model learn the knowledge distillated by the teacher model.

**Strengths:**

1. This paper proposed to capture topological and semantic information of nodes.
2. The experimental results show that the proposed method outperform all baseline methods across multiple datasets in two tasks.

**Weaknesses:**

1. There are many typos in the paper.
- Please add the whitespace between the text and the corresponding citation.
- What is the $L_{supervised}$ in equation 17? Should it be $L_{bpr}$?
- In equation 10, the objective function is the summation over N different nodes. However, in figure 1, the objective function is the summation over C different node types. The authors should correct the mismatch between these two objective functions.
- In equation 16,  $Z_v^{s}td$ should be $Z_v^{std}$, $Z_v^{t}ch$ should be $Z_v^{tch}$, $Z_v^{s}$ should be $Z_v^{std}$ and $Z_v^{t}$ should be $Z_v^{tch}$.

2. The motivation of this paper is not appealing. Many existing methods have shown the great performance in the limited label scenarios. Besides, the pretrained models aims to address this issue. What's the advantage of the proposed method over the existing methods?

3. In the experiment, it's better to highlight the method with the best performance.

4. In the introduction, the authors mention that of main issue of current supervised learning method is that their performance is highly dependent on having access to large amounts of labeled data, which is the key motivation of the proposed method. However, in both node classification and link prediction task in the experiment, the training datasets is 60% (80%), which cannot reflect the performance of the proposed method in the limited label scenario. It's recommended to conduct the experiments in a limited label scenario, e.g. 5% or 1% labeled nodes in node classification problem.

5. What's the time complexity of the proposed method compared with the baseline methods?

**Questions:**

1. What is the input node feature for the student model? What is the $Z_j$ in equation 11?

2. What's the meaning of the second term in equation 13? Can you further explain on this term?

3. Why do you incorporate the representation learned by teacher model in the final prediction in equation 14? If so, what's the point of distilling the knowledge from teacher model to student model? Why don't you just train the teacher model for the downstream task? Could you provide the intuition and the reason for that? In addition, in the summarization of contribution, the authors mention that the knowledge distillation mechanism balances performance and computational efficiency. If teacher model is used for final prediction, then how do you ensure the computational efficiency?

---

> ### Author Response · Authors · 2024-12-02
>
> Dear Reviewer K8FS
>
> I am following up on the revisions and responses I submitted regarding the comments and concerns raised during the review of my paper. I have addressed all the points raised and made significant improvements in the latest version of the manuscript. However, I have not yet received feedback or acknowledgment in the rebuttal section.
>
> Could you kindly confirm whether you have received my response? If so, I would appreciate it if you could consider the improvements made when evaluating the paper. As today is the final day for submitting a rebuttal, I am quite concerned. Your confirmation and feedback would be highly appreciated.
>
> Thank you for your time and consideration.

---

> ### Author Response · Authors · 2024-12-03
> **Urgent request for reevaluation of our revised submission before the deadline**
>
> Dear Reviewer,
>
> Thank you once again for your valuable feedback on my submission.
> We have carefully addressed all your comments in the revised version and submitted them for your consideration. As the $\textbf{final deadline}$ for reevaluation is in $\textbf{less than three hours}$, We kindly request you to \$\textbf{review the updated paper and provide your final feedback}$. I truly appreciate your time and effort.

---

### Note · Authors · 2025-09-22

I have read and agree with the venue's withdrawal policy on behalf of myself and my co-authors.

---

### Meta-Review · Area_Chair_oDMC · 2024-12-21

**Metareview:**

The paper presents a new knowledge distillation framework called KD-HGRL for heterogeneous graph representation learning. The problem is popular and interesting. Experimental results show that the proposed method outperforms all baseline methods. Many issues need to be addressed, mainly including insufficient experiments, incremental novelty, unclear experimental settings, and missing baseline comparisons. Reviewers are negative about this work.

**Additional Comments On Reviewer Discussion:**

No discussion is necessary as all give negative scores.

---

### Decision · Program_Chairs · 2025-01-22

Reject